# Comprehensive Geriatric Assessment for Frail Older People in Swedish Acute Care Settings (CGA-Swed): A Randomised Controlled Study

**DOI:** 10.3390/geriatrics5010005

**Published:** 2020-01-24

**Authors:** Katarina Wilhelmson, Isabelle Andersson Hammar, Anna Ehrenberg, Johan Niklasson, Jeanette Eckerblad, Niklas Ekerstad, Theresa Westgård, Eva Holmgren, N. David Åberg, Synneve Dahlin Ivanoff

**Affiliations:** 1Department of Health and Rehabilitation, Institute of Neuroscience and Physiology, Sahlgrenska Academy, University of Gothenburg, 405 30 Gothenburg, Sweden; isabelle.a-h@neuro.gu.se (I.A.H.); theresa.westgard@neuro.gu.se (T.W.); eva.holmgren@neuro.gu.se (E.H.); synneve.dahlin-ivanoff@neuro.gu.se (S.D.I.); 2Region Västra Götaland, Sahlgrenska University Hospital, Department of Acute Medicine and Geriatrics, 413 45 Gothenburg, Sweden; david.aberg@medic.gu.se; 3Centre for Aging and Health—AgeCap, University of Gothenburg, 405 30 Gothenburg, Sweden; 4School of Education, Health and Social Studies, Dalarna University, 791 31 Falun, Sweden; aeh@du.se; 5Department of Community Medicine and Rehabilitation Geriatric Medicine, Sunderby Research Unit, Umeå University, 901 87 Umeå, Sweden; johan.niklasson@umu.se; 6Department of Neurobiology, Care Sciences and Society, Karolinska Institutet, 171 77 Solna, Sweden; jeanette.eckerblad@ki.se; 7Region Department of Medical and Health Sciences, Division of Health Care Analysis, Linköping University, 581 83 Linköping, Sweden; niklas.ekerstad@vgregion.se; 8Department of Research and Development, NU Hospital Group, 461 73 Trollhättan, Sweden; 9Department of Internal Medicine, Institute of Medicine, Sahlgrenska Academy, University of Gothenburg, 405 30 Gothenburg, Sweden

**Keywords:** frail older people, comprehensive geriatric assessment, activities of daily living, geriatric, hospital care

## Abstract

The aim of the study is to evaluate the effects of the Comprehensive Geriatric Assessment (CGA) for frail older people in Swedish acute hospital settings – the CGA-Swed study. In this study protocol, we present the study design, the intervention and the outcome measures as well as the baseline characteristics of the study participants. The study is a randomised controlled trial with an intervention group receiving the CGA and a control group receiving medical assessment without the CGA. Follow-ups were conducted after 1, 6 and 12 months, with dependence in activities of daily living (ADL) as the primary outcome measure. The study group consisted of frail older people (75 years and older) in need of acute medical hospital care. The study design, randomisation and process evaluation carried out were intended to ensure the quality of the study. Baseline data show that the randomisation was successful and that the sample included frail older people with high dependence in ADL and with a high comorbidity. The CGA contributed to early recognition of frail older people’s needs and ensured a care plan and follow-up. This study is expected to show positive effects on frail older people’s dependence in ADL, life satisfaction and satisfaction with health and social care.

## 1. Introduction

Even though health care in Sweden is one of the best worldwide [1], many frail older people do not receive appropriate health care. Today’s specialised acute care is poorly adapted to the comprehensive care needs of frail older people and, therefore, exposes them to avoidable risks, such as loss of functional capacities, resulting in unnecessary health and social care needs as well as increased mortality [2]. In addition to appropriate specialised care when needed, assessments that are both comprehensive and person-centred are required to provide satisfactory and appropriate care to older people with complex needs [3]. Frailty is a state of decreased reserve resistance to stressors as a result of cumulative decline across multiple physiological systems, causing vulnerability to different outcomes such as falls, hospitalisation, institutionalisation and mortality [3,4,5]. Frail older people are at risk of further deterioration if their needs are not acknowledged [3]. The prevalence of frailty increases with age and is associated with an elevated risk of adverse health outcomes. Within Europe, the overall prevalence of frailty for people 65 years and older is approximately 10% with the northern countries having lower prevalence than the southern. Sweden has among the lowest, approximately 5% [6,7]. 

Previous research has found that the Comprehensive Geriatric Assessment (CGA) in acute hospital care is beneficial for frail older patients [8,9,10,11] and might be cost effective [12,13,14]. The CGA adopts a multi-dimensional team approach for assessing medical, functional, psychosocial and environmental needs [15]. The goal is to identify needs and provide support to help older people to be as independent as possible in their daily living. Key components of CGA interventions include coordinated multi-disciplinary assessment; geriatric medicine competence; identification of medical, physical, social and psychological problems; and the formation of a plan for care including appropriate rehabilitation [10]. Other components associated with improved outcomes of CGA are the ability to directly implement treatment recommendations made by the multi-disciplinary team and long-term follow-up [10]. Another key feature is the identification of frail patients. The acute care of older patients currently often takes place in acute care settings with short lengths of stay [16], and the CGA requires time and staff. For efficient use of healthcare resources, it is therefore important to identify those who can benefit the most from such an assessment by screening for frailty [17]. According to consensus from an international expert group, all persons aged ≥70 years should be screened for frailty [3]. Screening for frailty at the emergency department has proven effective [18,19] for identifying those in need of a more comprehensive assessment.

The benefits of CGA are highlighted in systematic reviews by both the Swedish Agency for Health Technology Assessment and Assessment of Social Services (SBU) and the Cochrane Collaboration [8,9,10,11]. Positive effects of CGA for frail older patients have been shown in the form of improved functional status, increased ability to remain in own housing and fewer readmissions. These results are important both for the frail older person and for society at large, because the CGA increases the individual’s possibility to live independently in their own home and leads to a decreased need for in-hospital and institutional care. However, both reviews state that substantial knowledge gaps concerning the effects of the CGA remain due to the lack of recent studies as well as those evaluating the CGA using validated measures [8,10,11]. Studies on the CGA have been conducted in various countries with different healthcare systems and demographic profiles. The CGA is a complex intervention that is highly dependent upon the context in which it is used [16], and there are very few recent studies carried out within the Swedish healthcare system thus limiting generalisation to the Swedish context. 

Swedish healthcare has undergone dramatic changes over the last decades, resulting in decreased numbers of hospital beds and shorter hospital stays [20], especially evident in geriatric hospital care [21]. This has led to Sweden having the lowest per capita hospital bed rate in Europe [20]. Despite the benefits of the CGA, it is a rather unknown concept within Swedish hospital care [11]. Over the past few years, there has been an increased interest, and many geriatric wards have implemented this way of working. However, geriatric hospital care is unevenly distributed within Sweden with a higher density within the bigger cities and university hospitals [11]. The CGA, within a Swedish community setting, has recently proven successful in maintaining/improving independence in activities of daily living (ADL) [22] as has the CGA in outpatient care [23]. One recent controlled study of an acute CGA unit reported the positive effects on health-related quality of life and mortality without higher costs [24]. Besides these studies, recent studies within Swedish acute care are scarce. Other limitations of these reviews are that many of the studies are dated, have limited sample sizes and have other methodological flaws [8,9,10,11]. More knowledge is needed on the implementation, effectiveness and cost effectiveness of the CGA in modern acute care settings. There is also a need to further investigate the relationship between frailty and the CGA as pointed out in a recent umbrella review of the CGA by Parker et al. [25]. This review also states that patient-related outcomes of the CGA—such as health-related quality of life, well-being and participation—are scarcely reported.

Thus, there is still a need for studies on the CGA and interventions using the CGA to improve acute care for frail older people in Swedish health care settings. To meet this need, we designed the study “Comprehensive Geriatric Assessment for Frail Older People in Swedish Acute Care Settings (CGA-Swed): A Randomised Controlled Study” with the purpose of improving hospital care for frail older people by implementing the CGA in Swedish acute care and testing the effects of the CGA in a randomised controlled study (RCT). The intervention was planned and elaborated in collaboration between professionals and researchers. The study includes both quantitative and qualitative analyses of the effects of CGA and a process evaluation of the implementation process throughout the intervention period. It started with a pilot and feasibility study that showed that the intervention—the CGA—and the research procedures were feasible [26]. There were also results indicating that the CGA increased patient safety [26]. Qualitative interviews with frail older people receiving care with the CGA at the intervention ward showed that they felt respected as persons when they were enabled to understand, engage in communication and participate in decisions [27]. The process evaluation alongside the RCT will add to the knowledge on the implementation of the CGA.

### Aim and Research Questions

The aim of the study was to evaluate the effects of the CGA for frail older people in Swedish acute hospital settings. The study addresses the following research questions:

Can the Comprehensive Geriatric Assessment for frail older people in Swedish acute hospital settings:Maintain independence in activities of daily living, functional status, health-related quality of life and life satisfaction?Increase satisfaction with health care?Reduce hospital and primary health care consumption?

How feasible and acceptable are the study processes and procedures of the CGA from the perspective of care givers and older persons in Swedish settings?

Ethical approval was obtained for the study, ref. no: 4,899-15, Regional Ethical Review Board in Gothenburg. Trial Registration: ClinicalTrials.gov, NCT02773914. 

This paper presents the study design, the intervention, baseline characteristics and the outcome measures of the study in accordance with the recommendations for reporting pragmatic randomised controlled trials by CONSORT [28].

## 2. Material and Methods

### 2.1. Project Context

The intervention “Comprehensive Geriatric Assessment for frail older people in Swedish acute care settings” took place at Sahlgrenska University Hospital/Sahlgrenska in Gothenburg, Sweden. Gothenburg is the second largest city in Sweden, situated on the west coast. It had 556,000 inhabitants in the year 2016. Sahlgrenska University Hospital is the regional hospital for Gothenburg and the surrounding municipalities, serving a total of almost one million inhabitants in 2016. The percentage of the population of people aged 75 and over for this region was 7.4% in 2016, compared to 8.6% for all of Sweden [29].

### 2.2. Study Design

The study was a two-armed randomised controlled trial that started with a pilot study in 2016. The CGA is a complex intervention that influences clinicians’ cognitive processes, requires multi-disciplinary collaboration and organisation of healthcare and is highly dependent upon the context in which it is used [16]. An RCT conducted in isolation may not be sufficient to provide guidance for decision makers in healthcare on whether or not to implement research findings. In addition, it is of interest to explore why, for whom and under what circumstances the intervention works. Thus, there was a process evaluation alongside the complex intervention as recommended by the Medical Research Council [30]. The pilot study and the process evaluation provide insight into the function and consequences of the intervention—why it works or fails—and help assess the feasibility of the intervention and research procedures. They also provide data on the number of factors that need to be assessed to successfully monitor and evaluate outcomes. This was an important part of the design that will shed light on real-life conditions that may be challenging in implementing the CGA. The pilot study with the first 30 participants showed that both the intervention and the research procedures worked well [26].

### 2.3. Study Population

The study group included 155 older people who sought acute medical care at the emergency department during the period between March 2016 and December 2018. Inclusion criteria were that participants were to be ≥75 years of age, in need of acute in-hospital care and screened as frail according to the FRESH-screening instrument. This screening instrument was chosen, since it was already implemented in clinical use, has high sensitivity and specificity (81% and 80%) for screening for frailty in acute settings and has an excellent clinical value [19]. It consists of four questions regarding dependence in shopping, tiredness, fatigue and risk of falling. Two or more “yes” responses indicate frailty. Exclusion criteria were not being screened as frail according to the FRESH-screening instrument, being admitted through a “fast track” for direct admission to a designated ward (for predefined diagnosis such as stroke, acute myocardial infarction and hip fracture) and having an acute severe condition requiring a higher level of care (e.g., intensive care) than the intervention ward. Cognitive impairment was not an exclusion criterion, and if the participant could not give informed consent due to the fact of cognitive impairment (*n* = 11), informed consent was obtained from next of kin.

### 2.4. Intervention Group

The intervention was the ward working according to the Comprehensive Geriatric Assessment. Key components of the CGA are multi-disciplinary teamwork, use of a person-centred approach [31], comprehensive assessments, treatment and rehabilitation, discharge planning and follow-up (see Figure 1). The multi-disciplinary team consisted of a physician, registered nurse (RN), assistant nurse (AN), physiotherapist (PT) and occupational therapist (OT) as well as other team members, such as a social worker (SW) and dietician, if needed. A pharmacist would have been valuable to include in the team, but this is not common in Swedish hospital wards. The senior physicians at the intervention wards were all specialist in geriatrics. The team had primary and continuing responsibility for assessment, planning of hospital care and discharge. Assessments of medical status, self-assessed health, functional status, psychological status, social situation and environment (see Table 1) were administered to ensure a comprehensive evaluation of the health and life situation of the frail older patient. The team used a person-centred approach [31] to individualise the assessment. A team conference was held every weekday to promote the sharing of information, experiences and competences in order to individualise the care for each patient. 

The content of the CGA was adapted to local routines and experiences and was elaborated in collaboration between the researchers and those working in the clinical setting where the intervention took place. This was to ensure that the CGA would be both clinically acceptable and evidence based. All personal at the intervention ward received education (i.e., information and workshops) on the CGA prior to the start of the study. During the whole study period, the researchers had meetings with representatives from the different professionals in order to follow how the teams experienced working according to the CGA.

The CGA starts at the intervention ward and continues throughout the hospital stay. It is individualised and unique for each patient, based on the key components in Figure 1 and the assessments in Table 1, providing a comprehensive assessment tailored for each person.

### 2.5. Control Group

The control group received the usual acute hospital care, i.e., care given at an ordinary medical hospital ward without a specialised multi-disciplinary team approach and without the CGA. The assessments and care provided at the control wards were based on the acute problem/symptom that the patients had and did not include the comprehensive and person-centred approach to the health and life situation of the patient inherent in a CGA. The occupational therapist and the physiotherapist worked more as consultants, and the amount of resources from occupational therapists and physiotherapists were lower at the control wards compared with the intervention wards. At the control ward, they did not perform functional tests, assessments of social network and total disease burden on all frail older patients. The senior physician at the control wards were specialists in internal medicine and were not geriatricians.

### 2.6. Procedures of the Intervention Study

Older patients who were screened as frail and were in need of in-hospital care were invited to participate in the study during the stay in the Emergency Department (ED). Those who agreed to participate were then randomised into control or intervention groups. Randomisation was done by computer-generated numbers and assigned by one of the researchers using QuickCalcs at GraphPad [32]. The allocation was concealed in numbered opaque envelopes. When a patient consented to participate, the hospital bed coordinator opened the envelope and admitted the patient to the designated ward. The intervention group was admitted to the ward according to the CGA and the control group to a general acute medical ward.

Baseline measures were collected by a research assistant who gathered data from the frail older patient and/or from the medical records during the hospital stay. Six participants were discharged before the baseline interview was completed. For these participants, parts or the whole of the baseline data collection was carried out in the participants’ homes as soon as possible after discharge. One participant was discharged to a surgical ward at a different hospital before the baseline interview, suffered a stroke after surgery and, thus, was admitted to the stroke ward at the intervention hospital. For this participant, the baseline interview was conducted retrospectively at the same time as the one-month follow-up which was performed during the stay at the stroke ward.

Follow-up interviews were performed in the older person’s home at 1, 6 and 12 months after hospital discharge. The follow-ups were performed using a structured questionnaire including questions about demographic data and assessments of outcome measures. The interviews lasted for approximately 1.5 h. Proxy interviews were done if the participant had a cognitive impairment making them unable to participate in parts or the whole of the baseline/follow-up data collection. No proxy interview was necessary at baseline. However, for a few participants, a next of kin was present during parts of the interview, providing additional information for participants who had difficulties remembering. Next of kin were also present at many follow-up interviews, supplementing the information provided during the interviews. A few of the follow-ups required proxy interviews. Whenever possible, the same person performed all follow-ups for the same participant, minimising the number of people that a frail older participant needed to meet.

The participants could not be blinded to the ward they were being admitted to, but they were not aware at which ward the intervention took place. However, they might have realised if they were administered the CGA. The wards could not be blinded to allocation, because the ward was the allocation. Not all patients at the intervention and control wards were included in the study, and the staff at the wards was not informed which patients were included in the study. In this respect, the staff could be seen as blinded. However, they might have observed the research assistant performing the baseline data collection and thereby understood that the patient was included in the study. Thus, the staff could not be completely blinded to allocation. The researchers performing the baseline interviews could not be blinded. The plan was that the collection of baseline measures and the performance of follow-up interviews were to be carried out by different researchers in order to keep the researchers blinded at the follow-up. However, the participants may have revealed the allocation (i.e., whether administered the CGA or not) during the follow-up interviews, making the interviewers non-blinded. To address this, we added a variable at all follow-ups indicating whether or not the allocation had been revealed which also reflects whether or not the participant was aware of being administered the CGA or not.

Meetings with representatives of all staff categories were regularly held at the intervention ward throughout the entire intervention period. There was a steering group with representatives from the research group and the different clinical care levels (geriatrics, internal medicine and rehabilitation) that met regularly, starting with the planning of the study and lasting throughout the whole study period.

The researchers (occupational therapists, physiotherapists, registered nurses and physicians) performing the interviews and measurements at baseline and follow-ups were all trained in observing and assessing in accordance with the guidelines for the outcome measurements.

### 2.7. Outcome Measures

For an overview of outcome measures and time of measure, see Table 2.

#### 2.7.1. The Primary Outcome

Dependence in daily activities was measured using the ADL-staircase assessment [33] by combining both interviews and observations. It includes dependence in nine activities: cleaning, shopping, transportation, cooking, bathing, dressing, going to the toilet, transferring and feeding. Dependence was defined as a state in which another person is involved in the activity by giving personal or directive assistance. The sum of dependence in the nine activities of daily living is calculated, range 0–9, with a clinically significant change of ≥1 unit between baseline and follow-up. At baseline, personal ADL (PADL: bathing, dressing, going to the toilet, transferring and feeding) was inquired for both actual PADL status during the hospital stay and retrospectively for PADL before onset of the acute illness leading to the hospital admission. This was done because the acute illness often leads to a higher dependence in PADL.

#### 2.7.2. The Secondary Outcomes

Functional status was measured using the Timed Up and Go (TUG) test [34], the Berg Balance Scale [35], Gait Speed four-metre walking test [36] and Grip Strength with North Coast Dynamometer [37]. The TUG test measures the time for a person to rise from a chair, walk 3 m, turn around, walk back, and sit down again. It measures both static and dynamic balance. In this study, we defined a change of ≥4 s as a clinically significant difference between baseline and follow-up, with 3.6 s considered as the minimal detectable change for TUG test measurements [42]. For details on Berg Balance Scale, Gait Speed and Grip Strength, see Section 2.8. Cognition was measured using the Mini Mental State Examination [38], see Section 2.8.

Self-rated was measured by the question: “In general, would you say your health is”, with the response alternatives: excellent, very good, good, fair, and poor. Clinically significant difference was defined as ≥1 step in the response alternatives between baseline and follow-up. In addition, self-reported symptoms were measured using the Göteborg Quality of Life Instrument [39].

Life satisfaction was measured using the Fugl–Meyer–Lisat-11 Questionnaire [40] which includes 11 items concerning satisfaction with: life as a whole, work, financial situation, leisure, friends and acquaintances, sexual life, functional capacity, family life, partner relationship, physical health and psychological health. Response alternatives included: very dissatisfied, dissatisfied, rather dissatisfied, rather satisfied, satisfied and very satisfied. In the analysis, the responses to each question were dichotomised into satisfied (very satisfied and satisfied) or not satisfied (rather satisfied, rather dissatisfied, dissatisfied and very dissatisfied) as was done in the validation of the questionnaire [40]. The sum of items for which the respondent reported being satisfied were calculated, range 0–11, with a clinically significant change of ≥1 between baseline and follow-up.

Satisfaction with quality of care was measured by the participant’s agreement with six statements with a person-centred approach: “I feel that the care given during the hospital stay meets my needs”, “I feel that the care planning meeting before discharge was valuable”, “I was able to take part in the discussion of my needs in the care planning meeting”, “I feel that the actions planned equal my needs”, “I feel that the actions delivered equal my needs” and “I am satisfied with the hospital care”. The response alternatives were agree completely, agree partly, neither agree nor disagree, disagree, and disagree completely. An answer of agree completely or agree partly were considered as satisfied. These questions were only measured once (at 1 month follow-up) and were used as the difference between intervention and control groups in the proportion of participants being satisfied for each question at follow-up as has been done previously [43].

Outcomes concerning health–economic aspects are health and social care consumption. Data on health care consumption can be retrieved from the regional care databases, including in-hospital and outpatient care, visits to primary healthcare (physicians, physiotherapists, occupational therapists, nurses, and assistant nurses) and home visits by primary healthcare professionals. The number of readmissions, number of in-hospital days, time until first readmission and number of outpatient visits were calculated and compared between intervention and control group for 1 year after study enrolment. Social care consumption was measured by questions about help received for instrumental and personal ADL from the municipality, privately financed help, relatives, friends and/or other. Response alternatives included none, less than once a week, once a week or more and daily. Extent and frequency of help received was calculated and compared between intervention and control groups for 1 year after study enrolment. In addition, questions covered institutional care, such as nursing home, retirement home and sheltered housing, from which the number of days in institutional care were calculated and compared between intervention and control groups for 1 year after study enrolment. 

To add to this, we used the ICECAP-O [41,44], which measures capability in older people, for use in the economic evaluation of health and social care interventions. It focuses on well-being defined in a broader sense and covers five attributes: (1) attachment; (2) security; (3) role; (4) enjoyment; and (5) control. The respondent chose one of five statements for each attribute. (1) Attachment: “I can have all the love and friendship that I want”; “I can have a lot of the love and friendship that I want”; “I can have a little of the love and friendship that I want”; “I cannot have any of the love and friendship that I want”. (2) Security: “I can think about the future without any concern”; “I can think about the future with only a little concern”; “I can only think about the future with some concern”; “I can only think about the future with a lot of concern”. (3) Role: “I am able to do all the things that make me feel valued”; “I am able to do many of the things that make me feel valued”; “I am able to do a few of the things that make me feel valued”; “I am unable to do any of the things that make me feel valued”. (4) Enjoyment: “I can have all the enjoyment and pleasure that I want”; “I can have a lot of the enjoyment and pleasure that I want”; “I can have a little of the enjoyment and pleasure that I want”; “I cannot have any of the enjoyment and pleasure that I want”. (5) Control: “I am able to be completely independent”; “I am able to be independent in many things”; “I am able to be independent in a few things”; “I am unable to be at all independent”.

Mortality rates will be retrieved from the National Cause of Death Registry.

### 2.8. Measurement of Frailty Indicators

In this study, we used the following measurements and cut-off levels of frailty indicators:

*Weakness:* Reduced grip strength considered to be below lowest norm range for ages 80–84, 13 kg for women and 21 kg for men for the right hand and below 10 kg for women and 18 kg for men for the left hand, using a North Coast dynamometer [37].

*Fatigue:* Question from the Göteborg Quality of Life Instrument [39], answering “Yes” to the question “Have you suffered any general fatigue/tiredness over the last three months?”

*Weight loss:* Question from the Göteborg Quality of Life Instrument [39], answering “Yes” to the question “Have you suffered any weight loss over the last three months?”

*Reduced physical activity:* Taking 1–2 or less outdoor walks per week.

*Impaired balance:* The Berg Balance Scale [35,45,46], reduced balance defined as having a value of 47 or less.

*Reduced gait speed:* Walking four metres with a gait speed of 0.6 metres/second or slower [36].

*Visual impairment*: The KM chart (Konstantin Moutakis chart) [47], impaired vision defined as having a visual acuity of 0.5 or less.

*Impaired cognition:* The MMSE [38], impaired cognition defined as having a score below 25.

### 2.9. Statistical Analysis and Power Calculation

A power calculation was done based on the primary outcome variable, dependence in activities of daily living (range 0–9) with an assumed difference between the intervention and control groups of one dependence (i.e., dependent in one or more activities of daily living, a clinically relevant difference of importance to the individual as well as the caregiver) and a standard deviation of 2 in both groups. To detect a difference between the intervention and control groups with a two-sided test and with a significance level of α = 0.05 and 80% power, at least 64 participants were needed in each group. To take a potential loss to follow-up into account, a total of 150 persons (75 in the control group and 75 in the intervention group) were initially planned to be included. This was later revised to allow for a higher loss to follow-up (22%) with 78 + 78, equalling a total of 156 participants. The assumed loss to follow-up and the power calculation were based on previous research on frail older people in need of acute care [48].

Both descriptive and analytical statistics were used in order to compare groups and to analyse changes over time. Non-parametric statistics were used when ordinal data were analysed. Otherwise, parametric statistics were used. Besides descriptive statistics, the chi^2^ and Fisher’s two-tailed exact tests to test differences in the proportions among the groups were used. A value of *p* ≤ 0.05 (two-tailed) were considered significant. The analysis were made on the basis of the intention-to-treat principle, meaning that participants were analysed on the basis of the group to which they were initially randomised. Given the old age of the participants, a relatively high drop-out rate was inevitable. Simply analysing complete cases is not relevant and might lead to bias, especially since missing data would not be at random. Therefore, the approach of data imputation was the replacement of missing values with a value based on the median change of deterioration between baseline and follow-up of all who participated in the follow-up [20]. The reasons for this imputation method were that (1) the study sample (frail older people) was expected to deteriorate over time as a natural course of the ageing process and (2) deteriorated health often is a reason for not fulfilling the follow-ups. Worst-case change was imputed for those who died before follow-up.

### 2.10. Process Evaluation

The process evaluation aims to provide insight into what is inside “the black box”, i.e., shed light on the function and consequences of the intervention—why it works or fails. The process evaluation includes context, recruitment, reach, dose delivered and received and fidelity [49], targeting recruitment and collection of outcome measures during the hospital stay and after discharge. The process evaluation focuses on the following aspects in line with the recommendations of the Medical Research Council [30]:The intervention, the actual exposure and the experience of the participants;Evaluation of which components of the intervention contributed to its success or failure;Description of the conditions under which the intervention is successful/unsuccessful.

The evaluation includes in-depth qualitative interviews with eight to ten participants in the intervention group, focusing on their experiences of receiving care according to the CGA [27]. The experiences of the staff working with to the CGA are explored through focus group discussions in order to gain an understanding of the intervention and its significance as well as its implementation. This includes the respondent’s role in working according to the CGA, perceptions about the CGA and its significance and effectiveness and possibilities and challenges involved in working according to the CGA. The focus group methodology distinctly utilises the interaction among participants in order to collect data, encouraging them to clarify not only what they think but also how and why they think in a certain way. The method is suitable for collecting the views and experiences of a selected group and generating a broad knowledge and understanding [50]. In addition, medical records are reviewed to assure that the assessments in the CGA have been conducted as intended, what aspects of CGA has been performed and by whom (i.e., the actual exposure).

### 2.11. Economic Analysis

The first step in the health–economic evaluation was to conduct a cost-minimisation analysis that compared the total costs between the CGA and CONTROL during the full follow-up period using the healthcare consumption data (measured as the cost of all resources used) as described above. This showed the cost implications for the payers if implementing the CGA or CONTROL.

The second step in the health–economic evaluation was to conduct a cost-effectiveness analysis. The cost-effectiveness of the intervention (versus the control) was evaluated based on the incremental cost-effectiveness ratio (ICER): ICER = (CostCGA – CostCONTROL)/(EffectivenessCGA – EffectivenessCONTROL). The effectiveness measure was based on the score from the ICECAP-O which measures older people’s capability for use in economic evaluation [41,44]. The ICER can be interpreted as the cost per one-unit gain in full capability and can, as such, be compared to other interventions using the same outcome measure in order to evaluate the relative cost-effectiveness. Sensitivity analysis and confidence intervals were calculated based on the non-parametric bootstrap approach.

### 2.12. Time Plan of the Study

The inclusion began in March 2016 and was completed in December 2018. The intervention began when the participant was admitted to the ward and lasted until discharge from the hospital. The follow-ups after one year are planned to be completed in January 2020. See Table 3 for time plan for the study and follow-ups.

## 3. Results

### 3.1. Baseline Characteristics

The inclusion and randomisation were carried out by the hospital bed coordinators at the emergency departments, because they are always involved when a patient is admitted to a ward, and they are responsible for coordinating available hospital beds. They had no extra time for this task and presumably did not always remember to inform and ask eligible patients. In addition, in many cases it was not possible to randomise due to the shortage of hospital beds at the wards (as there had to be a vacant bed at both the intervention ward and the control ward to be able to randomise). We asked the hospital bed coordinators to monitor how many patients were eligible and how many declined to participate. Unfortunately, due to their high work load, they only monitored this for approximately half a year. Based on the monitoring that was conducted, we estimated that approximately 210 eligible patients were asked to participate, and 178 of those consented to participate resulting in an estimated participation rate of approximately 85%. For details regarding the number of participants receiving allocated intervention with baseline data and reasons for declining participation, see Figure 2.

The median age of the participants in the inclusion year was 87 years in the control and 87.5 years in the intervention group. There were no statistically significant differences concerning baseline characteristics, frailty indicators and ADL between the control and intervention groups, see Table 4, Table 5 and Table 6.

### 3.2. Process Evaluation during Inclusion Period

There was a period (approximately half a year) with very high work strain for the CGA staff during the inclusion. The ward had to open up ten additional hospital beds within a couple of days due to the shortage of beds available at the hospital. This led to the need of hiring staff that did not have the knowledge and experience of working according to the CGA. Therefore, it is probable that a majority of the participants in the intervention group during this period did not receive a full CGA. Through the medical record review, we will investigate the extent to which the CGA was documented for each participant in order to be able to estimate the completeness of the CGA actually received by each participant in the intervention group.

In addition, there have been many readmissions during the follow-up period, in many cases to a different ward than during the inclusion. Thus, several participants in the control group may have received care at the CGA ward during the year after inclusion. This also needs to be considered when analysing follow-up data.

## 4. Discussion

The study “CGA-Swed” was designed to evaluate the effects of the CGA for frail older people in Swedish acute hospital settings. The primary outcome was dependence in ADL, as this has been pointed out as an important aspect for frail older people in previous research [52,53]. Secondary outcomes include other important aspects and patient-related outcomes such as self-rated health, life satisfaction, satisfaction with care, health care consumption and cost-effectiveness. Another strength of the study was the process evaluation alongside the RCT, adding to the knowledge on the implementation of the CGA and, thus, filling a knowledge gap pointed out by the Cochrane reviews of the CGA [9,10].

The randomisation seems to have been successful, as the baseline characteristics were similar between the intervention and control groups with no statistically significant differences among the two groups. Unfortunately, we do not know how many eligible participants could have been asked to participate nor how many in fact were asked to participate. This limitation could be seen as a consequence of performing a complex intervention in “real life”, being dependent on clinical staff performing parts of the research process. In our study, we were dependent on the hospital bed coordinators asking eligible participants beyond their ordinary duties as hospital bed coordinators and with no extra time or reward given. Similarly, it was not possible for us to have a researcher designated for the inclusion, since this would have required too much time and resources. Our estimate of a participation rate of 85% may seem high but seems realistic to us after discussion with the hospital bed coordinators. However, some of the participants had not fully understood what they had consented to, as they were asked about participation when seeking care for an acute medical problem. There is a risk that some were so comforted by being admitted to a ward that they consented out of pure relief. However, the researcher doing the baseline interview repeated the information about the study and asked once again about consent to participate, and very few participants declined participation at this occasion.

The baseline characteristics show that our sample was frail and had a high morbidity and high degree of dependence in activities of daily living. This is not surprising, since they all were screened as frail and in need of acute in-hospital care. The prevalence of frailty is known to be high among patients in internal medicine wards and especially in geriatric wards [54]. Thus, we argue that our sample is representative for frail older hospital medical in-patients. However, this also indicates a risk of high mortality within the sample, as both frailty and comorbidity are risk factors for high mortality [3,55]. Already during the hospital stay, three participants died, and preliminary data shows a high mortality rate to the follow-ups, higher than we expected based on earlier research [22,48].

The implementation process evaluation alongside the RCT—aiming to provide insight into the function and consequences of the intervention, why it works or fails—is an important part of the design that will contribute with insights into real-life conditions that may be challenging in implementing the CGA. The logistics of the intervention and the research procedures were tested in the pilot study and were found feasible [26]. The major deviation from the plan was the prolonged inclusion period because of the lack of available hospital beds which has prevailed throughout the study. However, we were eventually able to include the first estimated sample size of 150 participants, lacking only one participant to reach the revised estimated sample size of 156 participants. The process evaluation is a strength of the study, enabling us to generate knowledge on the process of the implementation of the CGA which has been pointed out as a knowledge gap by the Cochrane reviews of the CGA [9,10]. The results of the implementation process evaluation are planned to be presented in a forthcoming paper. There were some obvious threats to the study during the inclusion period and the follow-up period that we already observed and which may hamper our ability to demonstrate positive outcomes. The period with high work strain at the intervention ward is very likely to have led to CGAs of both lower quantity and quality than what was planned for. The patient medical record review will help us identify those not receiving a full CGA and allow us to adjust the analysis accordingly. However, this will probably lead to lower ability to detect a true difference between the groups. In addition, the fact that participants in the control group might have been admitted to the intervention group during the follow-up period needs to be accounted for in the analysis. As this can make the sample in each group smaller, this might also lead to lower power.

Findings on how frail older persons experience receiving care according to the CGA have already been published elsewhere based on this study, showing that they experienced being seen as a person while being admitted to a CGA ward [27].

To a large extent, we used the same questionnaires, measurements, manuals and outcomes as in our previous studies “Elderly Persons in the Risk Zone” [56] and “Continuum of Care for Frail Elderly People” [48]. This gives us the opportunity to compare among the studies with different levels of frailty within the samples, with the sample in “Elderly Persons in the Risk Zone” being prefrail [56] and the sample in “Continuum of Care” being less frail [48] than the sample in the current study. The outcome measures for the studies were carefully selected to make sure that they are valid and reliable for the target group and covering different components and levels of frailty.

We planned to have different researchers doing the baseline interviews and the follow-ups so as to be able to keep the researcher blinded at the follow-ups. However, in many cases it has been and will be the same research assistant doing the follow-up interviews who also conducted the baseline interview. Thus, we have not been able to keep the researcher blinded at the follow-up. The research assistant has, however, conducted most of the baseline interviews (*n* = 155) and in most cases did not remember the group assignment. It can also be seen as a strength that the same person carries out most interviews and assessments, as this ensures that the questions and assessments are done similarly at all occasions, thus enhancing the reliability of the assessments. In addition, having the same person in the follow-ups minimises the number of persons the participant has to meet. This might increase the chance of the participant remaining in the study, as they already are familiar with the person asking to do the follow-up. Since we had a variable at the follow-up regarding whether or not the researcher was aware of the allocation or whether the participant had revealed the allocation during the interview, we will know to what extent the researcher was blinded. 

The intervention was planned in collaboration with representatives from the Department of Geriatrics (i.e., intervention ward), the Department of Medicine (i.e., control wards) and the Department of Occupational Therapy and Physiotherapy. Regular meetings were held, starting with the planning of the study and lasting throughout the intervention period to discuss the content of the intervention and the research procedures which enhanced the implementation and strengthened the study. Both the research group and the group of professionals carrying out the intervention are multi-professional which is important since the CGA implies multi-professional team collaboration.

In summary, this study evaluated the CGA in current Swedish acute hospital settings employing a randomised controlled design, adding to the knowledge of the effects of the CGA in today’s hospital care which is characterised by shorter hospital stays and fewer available hospital beds than before. The results are expected to optimise the implementation of future complex interventions and lead to the improvement of care, support and rehabilitation of frail older people with complex needs. The process evaluation, aiming to provide insights into the function and consequences of the intervention—why it works or fails—is an important part of the design that will contribute with insights into real life conditions that may be challenging in implementing the CGA. This study is expected to show positive effects on frail older people’s dependence in activities of daily living, an outcome that is important for both the person and society. In addition, the CGA has potential to increase the satisfaction with care, life satisfaction and self-rated health as well as prevent deterioration in functional status and to be cost effective.

## Figures and Tables

**Figure 1 geriatrics-05-00005-f001:**
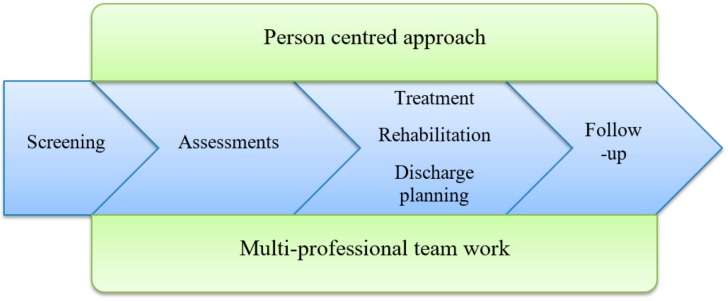
Key components of the Comprehensive Geriatric Assessment (CGA).

**Figure 2 geriatrics-05-00005-f002:**
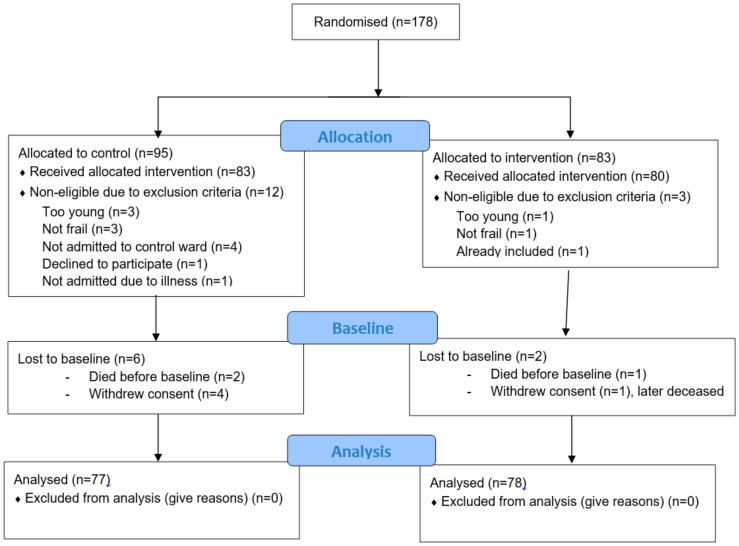
Flowchart of the three phases of the study implementing the “Comprehensive Geriatric Assessment in Swedish Acute Hospital Settings (CGA-Swed): A Randomised Controlled Study”, according to CONSORT [28].

**Table 1 geriatrics-05-00005-t001:** Assessments included in the CGA.

Domain	Assessment	Main Professional Responsible
Medical status	Illness burden/medical review	Physician
	Symptoms	Physician/RN
	Somatic status	Physician
	Pharmaceutical review	Physician
	Nutritional status	RN/Dietician
Self-assessed health	Self-assessed health	Physician/RN
Functional status	Activities of daily living	OT
	Physical function	PT
	Sight and hearing	RN/AN
Psychological status	Cognition	OT
	Depression	Physician
Social situation	Social network/informal support	RN/SW
	Formal support	RN/SW
	Financial support	RN/SW
Environment	Living conditions	RN/SW/AN
	Transports	RN/SW/AN
	Accessibility and assistive devices	OT/SW

RN = registered nurse; OT = occupational therapist; PT = physiotherapist; AN = assistant nurse; SW = social worker.

**Table 2 geriatrics-05-00005-t002:** Outcome measures and follow-ups.

Primary Outcomes	Measurement	Baseline	1 Month	6 Months	1 Year
Dependence	ADL-staircase [33]	X	X	X	X
**Secondary Outcomes**					
Functional status	Timed Up and Go [34]	X	X	X	X
	The Berg Balance Scale [35]	X			X
	Gait Speed 4 m [36]	X	X	X	X
	Grip Strength North Coast Dynamometer [37]	X	X	X	X
Cognition	Mini Mental State Examination (MMSE) [38]	X	X	X	X
Self-rated health	Questionnaire	X	X	X	X
	Symptoms: Göteborg Quality of Life Instrument [39]	X	X	X	X
Life satisfaction	Fugl–Meyer–Lisat-11 Questionnaire [40]	X	X	X	X
Satisfaction with quality of care	Questionnaire	X			
Health care consumption	Register Data				
Home help services	Questionnaire/Register Data	X	X	X	X
Capability	ICECAP-O [41]	X			X

Mortality	Register Data				

**Table 3 geriatrics-05-00005-t003:** Time plan for the study and follow-ups.

	Started	Completed	To Be Completed
Inclusion	March 2016	December 2018	
Baseline	March 2016	December 2018	
I month follow-up	April 2016	January 2019	
6 month follow-up	September 2016	July 2019	
12 month follow-up	March 2016		January 2020

**Table 4 geriatrics-05-00005-t004:** Baseline characteristics of the study population.

Characteristics	Control Group *N* = 77	Intervention Group *N* = 78	*p*-Value
Age, mean (range)	86.2 (76–98)	87.5 (75–101)	0.17
Female, % (*n*)	55.8 (43)	60.3 (47)	0.58
Living alone, % (*n*)	62.3 (48)	65.4 (51)	0.70
Academic education, % (*n*)	20.1 (16)	10.3 (8)	0.07
Good self-rated health, % (*n*) *	27.3 (21)	33.3 (26)	0.41
CIRS-G ≥ 3 in any category, % **	93.5 (72)	98.7 (77)	0.26
CIRS-G, median number of ratings 3–4 (range)	3 (0–9)	3 (0–7)	

* Excellent, very good or good. ** Cumulative Illness Rating Scale for Geriatrics. Rating 3 = severe/constant significant disability/uncontrollable chronic problem and rating 4 = extremely severe/immediate treatment required/end-organ failure/severe impairment in function [51].

**Table 5 geriatrics-05-00005-t005:** Frailty indicators.

Frailty Indicator	Control Group *N* = 77	Intervention group *N* = 78	*p*-Value
Fatigue, % (*n*) ^1^	90.9 (70)	87.2 (68)	0.46
Weight loss, % (*n*) ^2^	50.0 (38)	51.9 (40)	0.81
Weakness, % (*n*) ^3^	28.6 (22)	36.0 (27)	0.33
Reduced physical activity, % (*n*) ^4^	71.1 (54)	68.0 (51)	0.68
Impaired balance, % (*n*) ^5^	86.8 (66)	94.8 (73)	0.09
Reduced gait speed, % (*n*) ^6^	75.3 (58)	84.2 (64)	0.17
Visual impairment, % (*n*) ^7^	80.0 (60)	78.7 (59)	0.84
Impaired cognition, % (*n*) ^8^	48.1 (37)	52.0 (39)	0.63
Number of frailty indicators ^9^			
1, % (*n*)	1.3 (1)	0	0.88
2, % (*n*)	6.5 (5)	3.8 (3)	
3, % (*n*)	11.7 (9)	9.0 (7)	
4, % (*n*)	11.7 (9)	14.1 (11)	
5, % (*n*)	19.5 (15)	21.8 (17)	
6, % (*n*)	20.8 (16)	26.9 (21)	
7, % (*n*)	20.8 (16)	16.7 (13)	
8, % (*n*)	7.8 (6)	7.7 (6)	

**^1^** Answering “Yes” to the question “Have you suffered any general fatigue/tiredness over the last three months?” (Part of the Göteborg Quality of Life Instrument [39]). ^2^ Answering “Yes” to the question “Have you suffered any weight loss over the last three months?” (Part of the Göteborg Quality of Life Instrument [39]). Missing 1 in the control. ^3^ Reduced grip strength: below 13 kg for women and 21 kg for men for the right hand and below 10 kg for women and 18 kg for men for the left hand, using a North Coast dynamometer [37]. Missing 3 in the intervention. ^4^ Taking outdoor walks 1–2 times a week or less. Missing 1 in the control and 3 in the intervention. ^5^ Having a value of 47 or less on the Berg Balance Scale [35,45,46]. Missing 1 in the control and 1 in the intervention. ^6^ Walking four metres with a gait speed of 0.6 metres/second or slower [36]. Missing 2 in the control and 2 in the intervention. ^7^ Having a visual acuity of 0.5 or less using the KM chart [47]. Missing 2 in the control and 3 in the intervention. ^8^ Scoring below 25 on the Mini Mental State Examination (MMSE) [38]. Missing 3 in the intervention. ^9^ Missing information on 1–4 frailty indicators for 13 participants.

**Table 6 geriatrics-05-00005-t006:** ADL dependence at baseline.

Number of Dependences		Control Group*N* = 77	Intervention Group*N* = 78	*p*-Value
IADL dependence,	0	9.1 (7)	6.4 (5)	0.11
% (*n*)	1	19.5 (15)	7.7 (6)	
	2	18.2 (14)	14.1 (11)	
	3	20.8 (16)	21.8 (17)	
	4	32.5 (25)	50.0 (39)	
PADL dependence	0	61.0 (47)	52.6 (41)	0.83
before onset of	1	15.6 (12)	16.7 (13)	
illness, % (*n*)	2	7.8 (6)	12.8 (10)	
	3	7.8 (6)	7.7 (6)	
	4	5.2 (4)	5.1 (4)	
	5	2.6 (2)	5.1 (4)	
PADL dependence	0	31.2 (24)	23.1 (18)	0.38
during hospital	1	20.8 (16)	11.5 (9)	
stay, % (*n*)	2	9.1 (7)	12.8 (10)	
	3	9.1 (7)	10.3 (8)	
	4	24.7 (19)	33.3 (26)	
	5	5.2 (4)	9.0 (7)

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
