# Peer review of "Comprehensive Geriatric Assessment for Frail Older People in Swedish Acute Care Settings (CGA-Swed): A Randomised Controlled Study"

_geriatrics, 2020, doi:10.3390/geriatrics5010005_

Round 1

Reviewer 1 Report

The study investigate the effectiveness of CGA in frail acute patients in Sweden. The protocol is well written and describe RCT to test the effectiveness which is following a feasibility study that examined whether its feasible and acceptable to conduct the full trial. There is a huge evidence in the literature of the benefits of CGA globally which makes me still wonder the need for this study although the researchers explained their reasons and justification for the study. Personally I feel an implementation study that evaluate the feasibility and effectiveness of CGA within the Sweden Healthcare system might be more relevant and beneficial. However, If this protocol paper to be published a number of recommendations could improve the paper:

1- in the introduction, it would be useful to understand the national prevalence of frailty in Sweden compared to other countries. Also in the introduction, it would be useful to describe how and where CGA is currently implemented in clinical practice.

2- study sample: are you excluding patients who don't score frail on FRESH?can you add this to the exclusion criteria

can you explain why you chose this frailty screening tool not other more valid and commonly used ones such as PRISMA-7 or FRIED frailty phenotype?

3-Intervention group: the group is multidisciplinary but is pharmacists who have been proven to be highly valuable to CGA teams in terms of reviewing medications and making recommendations. so can you explain why a decision has been made to exclude pharmacists from CGA who might be the best clinician to conduct pharmaceutical review?

4- The researchers mentioned that the intervention will start at emergency department and continue during hospital admission. can you elaborate more on this? how would you record what aspect of CGA happens and by whom at each stage? how many clinicians in each speciality ?did the CGA team received any formal training to complete the different aspects of CGA?

5- the control group: it is unclear how participnats in the control group will be recruited and which wards will be recruited from? are control and intervention groups participants will be recruited from different ward who are looked after by different healthcare professionals? or are they recruited from the same wards? it is important to understand this and to explain how cross-contamination is avoided or dealt with. for meat the moment, it is unclear how participants in both groups are recruited, which wards, and who (clinicians) will see them?

6- in terms of grip strength measurement, the authors use very low cut off values of 13kg and 21kg for women and men, respectively. can you justify why using these cut off values despite that internationally the cut off values were set of at 20 and 30 from women and men!

7- page 14: line 494 remove the sentence "The baseline characteristics confirmed that our sample is a frail group with high comorbidity and high dependence in IADL" as it is repeating the first sentence in the paragraph

8- - the study design would have been better if used cluster-randomised trial which would have resolved many problems with allocation and blindiness

Author Response

Thank you very much for valuable comments on the manuscript.

We have now revised the manuscript according to the comments, and below you will find comments of the different points made. The alterations are highlighted in the downloaded manuscript using “track changes”.

Overall comment: ”There is a huge evidence in the literature of the benefits of CGA globally….”;  We agree with the reviewer that there are already much evidence for the benefits of CGA, but the evidence in current Swedish Healthcare is more limited, and the healthcare system has undergone many changes especially concerning geriatric hospital care. We hope that this is already clearly stated in the introduction.

“…an implementation study that evaluate the feasibility and effectiveness of CGA within the Sweden Healthcare system might be more relevant ant beneficial”. Yes, the implementation of CGA is of great interest. The process evaluation alongside the RCT aims to add to the knowledge on this. Information of the process evaluation is given under the heading 2.10 Process evaluation. We have added one sentence in the introduction section before the aims, lines 119-120, to further stress that the evaluation process will add to the knowledge on the implementation of CGA. We are also planning to present the results of the implementation process evaluation in a forthcoming paper. Information of this has been added lines 534-535 in the discussion.

Specific comments:

1.     We have added some information on the prevalence of frailty in Sweden in the introduction, first paragraph, lines 58-60. Information of implementation of CGA within Sweden has been added in the introduction lines 93-96.

2.     Yes, those not scoring frail on FRESH are excluded, since being screened as frail was an inclusion criteria. This has been added to the exclusion criteria under study population, lines 168-169. The reason for choosing the FRESH instrument has been added on lines 164-165.

3.     We agree that a pharmacist would be of high value. But the intervention ward had no pharmacist included in the team when the study started, and we could not influence this because this was the decision of the clinical organization. Pharmacist are not common in Swedish hospital wards, and information has been added on lines181-182.

4.     The CGA starts at the intervention ward. Thank you for pointing out the we have stated that it started at the ED. We have corrected the sentence on line 199. Information of senior physicians at the intervention ward all being specialist in geriatrics have been added on lines 182-183. As part of the process evaluation, medical records are reviewed to assure that the assessments in the CGA have been done as intended, and will thus give information on what aspects of CGA has happened and by whom. Information of already given under section 2.10 Process evaluation, lines 409-410 has been further elaborated. Information on training of personal has been added under section 2.4 Intervention group, lines 195-198.

5.     Participants in both groups were recruited at the ED. Information on this has been added under section 2.6 Procedures of the Intervention Study, line 217. After randomization, the intervention group were admitted to the ward working according to CGA and the control ward to an acute medical ward. Information on this has been added under section 2.6, lines 222-223. Information on different health care professionals at the intervention ward and the control wards are given under sections 2.4 Intervention group and 2.5 Control group. Additional information on the senior physician’s specialist competence have been added under both sections, lines 182-183 and 213-214. As the participants are cared for in different wards, the risk of cross-contamination is minor during the initial hospital stay. However, we have not been able to affect which ward they have been admitted to if there was need a readmission during follow-up, leading to a risk of cross-contamination during follow-up. Information of this is already given under section 3.2 Process evaluation during inclusion period, lines 485-488, and discussed in the discussion lines 541-544.

6.     The cut-offs for grip-strengths were chosen according to the lowest norm range scores given by the North Coast Hand Dynamometer for age 80-84 for men (21-44kg) and women (13-26kg). Additional information of this is added section 2.8 Measurement of frailty indicators, line 348. We acknowledge that this is lower than the usually used cut offs, and is a limitation to the study. On the other hand, it is the same as we have previously used in other studies, which makes it possible for us to compare to our previously performed research that has used the same questionnaires, measurements, manuals and outcomes as in the present study. This is discussed at lines 548-555.

7.     The sentence has been removed (lines 517-518 in the revised manuscript)

8.     We do not fully understand the reviewer’s comment on a cluster-randomised trial as a better design. There were no major problems with allocation, and we do not see how blindiness would be easier with cluster randomisation.

Other revisions made:  

We observed that we missed to give information on CIRS-G in table 4. Thus, we have added a footnote with this information and a reference.

The new information concerning reviewer 1’s point 1 concerning prevalence of frailty in Sweden needed additional references added. Therefore, the numbering of references has been revised.

Thank you very much for giving us the chance to further improve the manuscript, guided by your valuable comments.

Reviewer 2 Report

This is an excellent presentation of a research protocol that explains the study and methodology to be used. The topic of the comprehensive geriatric assessment use in hospital settings is highly relevant to this journal and readership. 

Intro is well written and thorough: Love the statement "Today’s specialised acute care is poorly adapted to the comprehensive care needs of frail older people and therefore exposes them to avoidable risks such as loss of functional capacities, resulting in unnecessary health and social care needs as well as increased mortality"

Section 2.4 - Not clear why this is italicized? 

The primary outcome and secondary outcome section headings should be more uniform. Pull out "Primary Outcome:"

Overall well done. 

Author Response

Thank you very much for valuable comments on the manuscript.

We have now revised the manuscript according to the comments made by the referees, and below you will find comments of the different points made. The alterations are highlighted in the downloaded manuscript using “track changes”.

Thank you for your positive comments.

Section 2.4: this is no longer italicized.

Primary outcome is now pulled out as a subheading.

Other revisions made:  

We observed that we missed to give information on CIRS-G in table 4. Thus, we have added a footnote with this information and a reference.

The new information concerning reviewer 1’s point 1 concerning prevalence of frailty in Sweden needed additional references added. Therefore, the numbering of references has been revised.